| Editor's Pick | Host-Microbial Interactions | Research Article

# Impact of host species on assembly, composition, and functional profiles of phycosphere microbiomes

Line Roager,[1] Paul J. Kempen,[2,3] Mikkel Bentzon-Tilia,[1] Eva C. Sonnenschein,[1] Lone Gram[1]

**ABSTRACT** Microalgal microbiomes play vital roles in the growth and health of their host, however, their composition and functions remain only partially characterized, especially across microalgal phyla. In this study, a natural seawater microbiome was introduced to three distinct, axenic species of microalgae, the haptophyte *Isochrysis galbana,* the chlorophyte *Tetraselmis suecica,* and the diatom *Conticribra weissflogii* (previously *Thalassiosira*), and its divergence and assembly under constant illumination was monitored over 49 days using 16S rRNA amplicon and metagenomic analyses. The microbiomes had a high degree of host specificity in terms of taxonomic composition and potential functions, including CAZymes profiles. Rhodobacteraceae and Flavobacteriaceae families were abundant across all microalgal hosts, but *I. galbana* microbiomes diverged further from *T. suecica* and *C. weissflogii* microbiomes. *I. galbana* microbiomes had a much higher relative abundance of Flavobacteriaceae, whereas the two other algal microbiomes had higher relative abundances of Rhodobacteraceae. This could be due to the bacterivorous mixotrophic nature of *I. galbana* affecting the carbohydrate composition available to the microbiomes, which was supported by the CAZymes profile of *I. galbana* microbiomes diverging further from those of *T. suecica* and *C. weissflogii* microbiomes. Finally, the presence of denitrification and other anaerobic pathways was found exclusively in the microbiomes of *C. weissflogii,* which we speculate could be a result of anoxic microenvironments forming in aggregates formed by this diatom during the experiment. These results underline the significant role of the microalgal host species on microbiome composition and functional profiles along with other factors, such as the trophic mode of the microalgal host.

**IMPORTANCE** As the main primary producers of the oceans, microalgae serve as cornerstones of the ecosystems they are part of. Additionally, they are increasingly used for biotechnological purposes such as the production of nutraceuticals, pigments, and antioxidants. Since the bacterial microbiomes of microalgae can affect their hosts in beneficial and detrimental ways, understanding these microbiomes is crucial to both the ecological and applied roles of microalgae. The present study advances the understanding of microalgal microbiome assembly, composition, and functionality across microalgal phyla, which may inform the modeling and engineering of microalgal microbiomes for biotechnological purposes.

**KEYWORDS** microalgae, phycosphere, metataxonomic analyses, metagenomics, microbiome, *Tetraselmis suecica*, *Isochrysis galbana*, *Conticribra weissflogii*

Microalgae are fundamental to aquatic ecosystems and are the main primary producers of the oceans. Therefore, microalgae have become popular model systems in the field of microbial ecology allowing studies of host-microbiome interactions (1–5). Also, they are important components of many industrial processes, for

Address correspondence to Lone Gram, gram@bio.dtu.dk.

The authors declare no conflict of interest.

See the funding table on p. 15.

instance, as live feed in aquaculture (6–9) and as cell factories in the production of biofuels, nutraceuticals, and natural pigments (10, 11).

As is the case for higher eukaryotes, microalgae host a microbiome in their vicinity. The nutrient-rich boundary layer surrounding microalgal cells has been coined the phycosphere; an aquatic analogue to the plant rhizosphere (12). Several marine bacteria utilize chemotaxis for the detection of compounds released by phytoplankton (12–17), and it has been proposed that direct interactions between microalgae and bacteria take place in the phycosphere (18). Beneficial as well as detrimental effects of phycosphere bacteria to host microalgae have previously been observed. For instance, signaling molecules such as indole-3-acetic acid or its precursor tryptophan can be released by bacteria to induce microalgal growth (3, 5, 19), or certain host-specific phycosphere bacteria can protect the host from antagonistic effects of algicidal bacteria (20–22). Phycosphere bacteria may also cause cell lysis of microalgae by proteases or the algicidal roseobacticides (23, 24). Some interactions may even change character based on specific cues in a beneficial-then-detrimental manner, for instance, the interaction between *Phaeobacter gallaeciensis* and the coccolithophore *Emiliania huxleyi*, where the bacterium provides auxins to the algal host in initial bloom phases, but then switches to production of roseobacticides during algal senescence (24).

Most studies investigating microalgal-bacterial interactions have been conducted in reduced laboratory-based systems with one algal host species and one bacterial species (5, 22, 25–28). However, recent studies have demonstrated that the same microalgal host can induce very different responses in different bacterial species and vice versa (4, 28–30). Hence, ecological implications of the interplay between microalgae and phycosphere bacteria may not be directly deduced from simple co-cultures of one microalgal host and one bacterial species. Instead, studying the complex microbiomes associated with microalgae could aid in elucidating the ecological impact of microalgae and their microbiomes.

While the importance of microalgal microbiomes is increasingly being recognized, factors determining their establishment and development are still not fully understood. The microbiomes of diatoms, the most dominant group of microalgae in the oceans, are host-specific (31, 32), and in some cases even strain-specific (33). Other phylogenetic groups of microalgae are less studied in terms of microbiome compositions and host specificity, but a recent study found the assembly of microbiomes of four different freshwater chlorophytes to be highly host-specific (34). Even in alpine snow algae, host specificity of microbiomes has been observed (35). However, many studies investigating microalgal microbiomes have not investigated the assembly over time or focused on one or a few closely related microalgal hosts (36–38).

While the aforementioned studies point toward a high degree of host specificity of microalgal microbiomes, some bacterial lineages such as Flavobacteriales and Rhodobacterales are repeatedly found in association with phytoplankton blooms (39–42). Despite large variations in dominating phytoplankton species and environmental conditions, this has led to the proposition of the existence of "archetypal phytoplankton-associated bacteria" (40). With the common functional roles of microalgae as primary producers and thus regulators of carbon cycling in the oceans, it seems likely that the nutrient-rich environment of the phycosphere could harbor some core microbiome members regardless of host species and phylum. Some bacteria might be "opportunistic" in the sense that they will always take advantage of a nutrient-rich environment regardless of which host microalga provides it. In addition to host specificity, the broader functions of microalgal microbiomes are largely unknown. In another host-microbiome system, the human microbiome project, the functionality of an array of human-associated microbiomes was found to be very similar even across taxonomically distinct microbiomes (43, 44). This may also be the case for microalgal microbiomes but currently remains unknown.

The purpose of the present study was to determine how microbiomes assemble and develop in association with different species of host microalgae over time, starting from

the same natural seawater microbiome. Using 16S rRNA amplicon and metagenomic sequencing and analyses, assembly dynamics and functionality of the microbiomes of three distantly related species of marine microalgae were investigated: the hapto-phyte *Isochrysis galbana,* the chlorophyte *Tetraselmis suecica,* and the diatom *Conticribra weissflogii*. All three species are important as live feed in the aquaculture industry (6–8), where the microalgal microbiomes are suspected to be potential vectors for pathogens detrimental to the aquaculture livestock (45, 46). Moreover, as biotechnological interest in the production of omega-3 polyunsaturated fatty acids (PUFAs) is increasing, these microalgal species have potential as producers of high amounts of the PUFAs docosa-hexaenoic acid and eicosapentaenoic acid (7, 9, 47–49). With microalgal microbiomes having beneficial and in some cases, detrimental effects on algal growth and health, a better understanding of the functionality, composition, and assembly of microalgal microbiomes is necessary to be able to engineer microalgal microbiomes for the benefit of any of these biotechnological applications in the future.

## MATERIALS AND METHODS

### Microalgal cultures

Axenic cultures of *I. galbana* CCMP1323, *C. weissflogii* CCMP1336, and *T. suecica* CCAP 66/4 were obtained from the Bigelow National Center for Marine Algae and Microbiota (US, Maine) and the Culture Collection of Algae and Protozoa (UK, Scotland). Cultures were maintained routinely at 18°C and ~50 µmol photons $m^{-2}$ $s^{-1}$ in f/2 medium (50, 51) based on 3% Instant Ocean (IO; Aquarium Systems) without silica for *I. galbana* CCMP1323 and *T. suecica* CCAP 66/4 (f/2 − Si 3% IO), and with silica for *C. weissflogii* CCMP1336 (f/2 + Si 3% IO). Cultures were confirmed to be axenic by plating 100 µL of each microalgal culture on marine agar (Difco, MA), incubating at 25°C, and checking for visible colony growth after 7 and 14 days. Additionally, a microscopy check of axenicity was performed by phase-contrast microscopy using an Olympus BX51 microscope and 1,000× magnification.

### Collection of seawater and preparation of microbiome inoculum

Twenty liters of natural seawater was collected in November 2021 in Øresund between Denmark and Sweden (N 56.066167, E 12.648250) with a 12 L Niskin bottle at 12 m depth. One liter of untreated seawater was filtered on 0.22 µm Sterivex polyethersulfone (PES) membrane filters in quadruplicates for DNA extraction using a peristaltic pump. Additionally, 50 mL seawater was used for chemical analysis of organic carbon by non-volatile organic carbon analysis, analysis of P, S, Fe, Mg, Na, Ca, and Si by inductively coupled plasma-optical emission spectrometry, and analysis of $NO_3^-$, $NH_4^+$, $NO_2^-$, $PO_4^{3-}$, $SO_4^{2-}$, and $Cl^-$ by ion chromatography. The remaining seawater was filtered through 3 µm polycarbonate (PC) filters as soon as possible after collection (i.e., within a few hours) to remove eukaryotic plankton and particulate material and is from here on referred to as the microbiome inoculum. From the microbiome inoculum, 10 mL was filtered onto 0.22 µm Sterivex PES filters in quadruplicate for DNA extraction.

### Enumeration of bacteria in the microbiome inoculum

For immediate enumeration of bacteria in the microbiome inoculum, 5 mL was filtered onto a 0.02 µm PC filter with a 1 µm methylcellulose ester support filter as described previously (52, 53). The filter was stained with a 0.35% SYBR Gold solution for 15 min in the dark and the staining solution was washed off. After drying, the filter was mounted on a glass slide with 43.5% glycerol and 0.1% p-phenylene-diamine in 50% phosphate-buffered saline, and the sample was analyzed on an Olympus BX51 fluorescence microscope with a 460–495-nm excitation filter, and emission filter > 510 nm. Ten fields of vision were chosen randomly and counted using ImageJ (54).

## Microbiome assembly and sampling

Cultures of *I. galbana* CCMP1323, *T. suecica* CCAP 66/4, and *C. weissflogii* CCMP1336 with microbiome inoculum were prepared in biological quadruplicates by adding axenic microalgal pre-cultures at 10% to the microbiome inoculum with the addition of nutrients, vitamins, and trace metals in amounts corresponding to f/2 ± Si medium. Axenic controls were also prepared of all microalgal hosts in sterile-filtered seawater with added nutrients, vitamins, and trace metals corresponding to f/2 ± Si medium. All cultures were kept at 18°C at constant illumination ~50 µmol photons $m^{-2}$ $s^{-1}$ for 7 days before re-inoculation (10-fold dilution) into fresh f/2 ± Si 3% IO. Cultures were transferred every 7 days for 7 weeks, a total of 49 days. Samples for DNA extraction were taken from cultures with microbiomes at day 0 (i.e., the day the cultures were prepared), 7, 14, 28, and 49 by filtering 10 mL (50 mL for day 0) grown culture through 5 µm PC filters to capture of the fraction of bacteria in the microbiome attached (A fraction) to the microalgae, and sequentially filtered through 0.2 µm PC filters to capture the free-living (FL) fraction of the microbiome. Additionally, samples for enumeration of microalgae were taken both before and after each dilution as well as on day 0 by sampling from all cultures and fixing 1 mL of culture with 1% formaldehyde. Samples were then stored at 4°C until quantification by flow cytometry (MACSQuant VYB, Miltenyi Biotec). This was done by running 100 µL of unstained sample at a flow rate of 25 µL $min^{-1}$ and using forward scatter (FSC) as a data collection trigger. Both *I. galbana* and *T. suecica* cells were quantified based on FSC and chlorophyll autofluorescence (561 nm excitation/655–730 nm emission filter), whereas *C. weissflogii* cells were counted by microscopy using a Neubauer-improved chamber. The resulting data were analyzed using the MACQuantify Software. Cultivable bacterial counts in microbiome cultures were determined after preparing a 10-fold dilution series in 3% IO and plating on MA, and axenicity checks of axenic control cultures were conducted as described above at each transfer of cultures.

## Microbiome characterization by 16S rRNA gene amplicon sequencing

For DNA extraction, PC filters were cut into smaller pieces using sterile scissors and submerged in 1 mL lysis buffer (400 mM NaCl, 750 sucrose, 20 mM EDTA, 50 mM Tris-HCl, 1 mg $mL^{-1}$ lysozyme, pH 8.5). Samples were incubated at 37°C for 30 min before the addition of proteinase K (final concentration 100 µg $mL^{-1}$) and sodium dodecyl sulfate (SDS, final concentration 1%). Samples were lysed overnight at 55°C and 300 rpm. The lysate was added to Maxwell RSC Cartridges, and DNA was extracted automatically using a Maxwell Instrument (Promega) into the elution buffer provided in the Maxwell RSC Blood DNA Kit. Upon extraction, DNA concentrations were measured using a Qubit 2.0 Fluorometer and the high sensitivity (HS) assay kit. For characterization of the microbiome, amplicon sequencing of the V3–V4 region of the 16S rRNA gene was performed using barcoded versions of the primer set 341 f (5′-CCTACGGGNGGCWGCAG-3′) and 805 r (5′-GACTACHVGGGTATCTAATCC-3′) (55). Amplification was performed by PCR on a Bio-Rad T100 Thermal Cycler using TEMPase Hot Start 2× Master Mix (Ampliqon), 10 µM forward primer, 10 µM reverse primer, and 1 µL template DNA. The PCR program consisted of an initial denaturation at 95°C for 15 min, then 30 cycles of denaturation at 95°C for 30 s, annealing at 60°C for 30 s, and elongation at 72°C for 30 s, and a final elongation step at 72°C for 5 min. Selected samples with low amounts of extracted DNA were subjected to the same PCR program but with 35 cycles (Table S1). Subsequently, amplicons were purified using the AMPure XP reagent (Beckman Coulter) and the appertaining protocol for PCR cleanup. Negative controls were made from PC filters that were subjected to the entire extraction, amplification, and cleanup pipeline. Four controls were subjected to the PCR program with 30 cycles while four other controls were subjected to the 35-cycle PCR program. The purified products were quantified using the HS assay kit and a Qubit 2.0 Fluorometer and pooled in equal amounts before sequencing on an Illumina Novaseq 6000 with paired-end 250 bp reads (Novogene).

## Analysis of 16S rRNA gene amplicon sequencing data

Sequencing data were analyzed using QIIME 2 2022.2 (56) and R (v. 4.2.2) (57). The raw reads were de-multiplexed and barcodes and primers were removed using the cutadapt plugin of QIIME 2 (58). Subsequently, reads were denoised, filtered, dereplicated, and merged using the DADA2 plugin (59). An amplicon sequencing varint (ASV) table was also produced with DADA2 using pseudo-pooling of samples for ASV inference. Taxonomy of ASVs was assigned using the feature-classifier plugin (60) and a classifier trained with the 341 f/805 r primer pair mentioned above on the Silva 138 database (61) with 99% sequence similarity. ASVs classified as chloroplasts or mitochondria were filtered out before further analysis. Using R and the decontam package (62), the negative controls were used to remove reads that were most likely contaminated from samples. For further analysis, data were square root transformed and normalized using Wisconsin standardization. Subsequently, the vegan package (63) was used to perform non-metric multidimensional scaling (NMDS) and permutational analyses of variance (PERMANOVA). Using the ampvis2 package (64), alpha diversity metrics were computed. Data visualizations and handling were done using the tidyverse package and the packages therein (65). Differentially abundant ASVs associated with a certain host alga and fraction were determined using the ANCOM-BC package (66).

## Shotgun metagenomics of assembled microbiomes

Forty-nine days after the introduction of the microbiome inoculum to microalgal cultures, 90 mL of the microalgal cultures with assembled microbiomes were filtered onto Sterivex 0.22 µm PES filters using a peristaltic pump. Filters were stored at −20℃ until further processing. DNA was extracted using a phenol-chloroform protocol adapted from Boström et al. (67). In short, Sterivex cases were opened and filters were cut into smaller pieces using sterile scalpels. Cells on the filters were then lysed enzymatically for 30 min at 37℃ in a sucrose lysis buffer (400 mM NaCl, 750 mM sucrose, 20 mM EDTA, 50 mM Tris-HCl, 1 mg mL$^{-1}$ lysozyme, pH 8.5). Proteinase K and SDS were added to final concentrations of 100 µg mL$^{-1}$ and 1%, respectively, and samples were incubated at 55℃ overnight with agitation at 300 rpm. Subsequently, phenol:chloroform:isoamylalcohol (25:24:1, vol/vol/vol) was added to samples in an amount equal to that of the samples, samples were mixed thoroughly and centrifuged for 5 min at 4℃ and 20,000 × $g$. The supernatant was transferred and chloroform:isoamylalcohol (24:1, vol/vol) was added in an amount equal to the sample. Samples were mixed thoroughly and centrifuged for 5 min at 4℃ and 20,000 × $g$, and supernatant was transferred. A 0.1 vol of sodium acetate (3 M) was then added to the samples along with 0.6 vol of cold isopropanol, samples were gently mixed and incubated at −20℃ for 2 h. Finally, samples were centrifuged for 20 min at 4℃ and 20,000 × $g$, the supernatant discarded and the pellet washed with cold 70% ethanol. Samples were centrifuged for 20 min at 4℃ and 20,000 × $g$, the supernatant discarded and the pellet air-dried before resuspension in TE buffer at 56℃. The extracted DNA of microalgae and their assembled microbiomes was purified by magnetic bead purification before library prep and shotgun metagenomics sequencing on an Illumina Novaseq 6000 instrument with 150 bp paired-end reads (Novogene).

Initial handling of sequencing data was performed using ATLAS v. 2.16.2 (68). In short, raw reads were quality filtered, trimmed, and deduplicated, and host DNA was removed by mapping against in-house available host genomes of *I. galbana* and *T. suecica*, and against the *C. weissflogii* genome publicly available with ID CNA0030133 at the China National Gene Bank database (69). Reads were then merged and assembled into contigs using MEGAHIT (70). Open reading frames were identified and translated using prodigal (71). Using the dbcan4 tool (72, 73), an analysis of potential carbohydrate-active enzymes (CAZymes) was performed in addition to functional annotation as assigned by the Kyoto Encyclopedia of Genes and Genomes (KEGG) Orthologs (KOs) using the GhostKOALA tool against the prokaryotes and viruses database (74, 75). CAZymes detected with less than two methods (of HMMER, dbcan substrate database, and DIAMOND BLAST) were filtered out before downstream handling. The resulting output data were handled and

visualized using KEGGdecoder and in R 4.3.1 using packages vegan, tidyverse, ampvis2, and phyloseq (63–65, 76, 77).

## Scanning electron microscopy of diatom cells with attached bacteria

A volume of 120 µL of sample from day 49 of *C. weissflogii* cultures with microbiomes was mixed with 40 µL of 8% glutaraldehyde in water and left for 30 min at room temperature before being stored overnight at 4°C. The following morning, 80 µL of the sample was placed on two poly-L-lysine coated glass coverslips and allowed to adhere for 30 min before being rinsed in milliQ water three times. Samples were then stained in 1% osmium tetroxide for 60 min followed by two rinses in milliQ water. Coverslips were then dehydrated in increasing concentrations of ethanol at 15 min intervals; 50%, 70%, 95%, 100%, and 100%, before being critical point dried using a Leica EM CPD300. Following critical point drying, coverslips were mounted on aluminum scanning electron microscopy (SEM) stubs and coated with 5 nm of gold using a Quorum 150T sputter coater. Samples were imaged using a Thermofisher Quanta FEG 200 SEM operated at an accelerating voltage of 5 kV.

## RESULTS

### Seawater inoculum, cell abundances, and attachment

To study the association of natural marine microbiota with three microalgal host species, seawater was sampled from Øresund. At the time of sampling, the seawater had a temperature of 12°C, a salinity of 24 PSU, and contained per liter seawater; S 603 mg, Mg 810 mg, Na 5,838 mg, Ca 260 mg, $Cl^-$ 8,679 mg, $SO_4^{2-}$ 1,567 mg, $PO_4^{3-}$ 20 µg, $NO_3^-$ 13 µg, $NH_4^+$ 136 µg, and total organic carbon 4,473 µg. P, Fe, and Si concentrations were below the detection level of 0.5 mg $L^{-1}$ seawater. After 3 µm filtration of the seawater, the microbiome inoculum contained $5.37 \times 10^6$ bacterial cells $mL^{-1}$ as assessed by microscopy. After the addition of the inoculum to the axenic microalgal hosts, the cultivable counts of the assembling microbiomes as assessed by plating on marine agar plates were not assessed on day 0, but on day 7 was at a mean of $1.14 \times 10^5 \pm 1.6 \times 10^4$ CFU $mL^{-1}$ for *I. galbana* and a mean of $3.45 \times 10^5 \pm 3.4 \times 10^5$ CFU $mL^{-1}$ for *T. suecica* cultures, and increased over time to $2.80 \times 10^6 \pm 2.2 \times 10^6$ CFU $mL^{-1}$ and $5.81 \times 10^6 \pm 4.4 \times 10^6$ CFU $mL^{-1}$, respectively, on day 49 (Fig. S1), indicating a possible selection for bacteria cultivable on marine agar in the assembling microbiomes. Cultivable counts of assembling microbiomes associated with *C. weissflogii* decreased slightly over time, starting at $4.17 \times 10^6 \pm 1.1 \times 10^6$ CFU $mL^{-1}$ on day 7 and decreasing to $1.25 \times 10^6 \pm 4.7 \times 10^5$ CFU $mL^{-1}$ on day 49 (Fig. S1).

At day 0, the microalgal cultures contained a mean of $5.00 \times 10^5 \pm 1.5 \times 10^5$ algal cells $mL^{-1}$ in *I. galbana* cultures, $1.13 \times 10^5 \pm 9.5 \times 10^4$ algal cells $mL^{-1}$ in *T. suecica* cultures, and $8.13 \times 10^3 \pm 4.0 \times 10^3$ algal cells $mL^{-1}$ in *C. weissflogii* cultures (Fig. S2). The algal cell densities remained in this realm after each dilution step throughout the experiment, however, cultures of *C. weissflogii* experienced a slight decrease in algal cell concentrations over time. Cultures with and without microbiome inoculum did not show a difference in growth rate for any of the microalgae (Fig. S2), however, flocculation of microalgal cells was observed in *C. weissflogii* cultures with microbiome inoculum throughout the experiment, while not observed in axenic cultures.

Through SEM, bacterial cells on the surface of *C. weissflogii* cells were observed along with unidentified substances, which we speculate are extracellular polymeric substances produced by the alga or by microbiome members. No such substances were observed through SEM imaging of axenic *C. weissflogii* cells (Fig. S3). Bacterial cells were especially observed, although not quantified, in areas where these substances were present on *C. weissflogii* surfaces (Fig. 1).

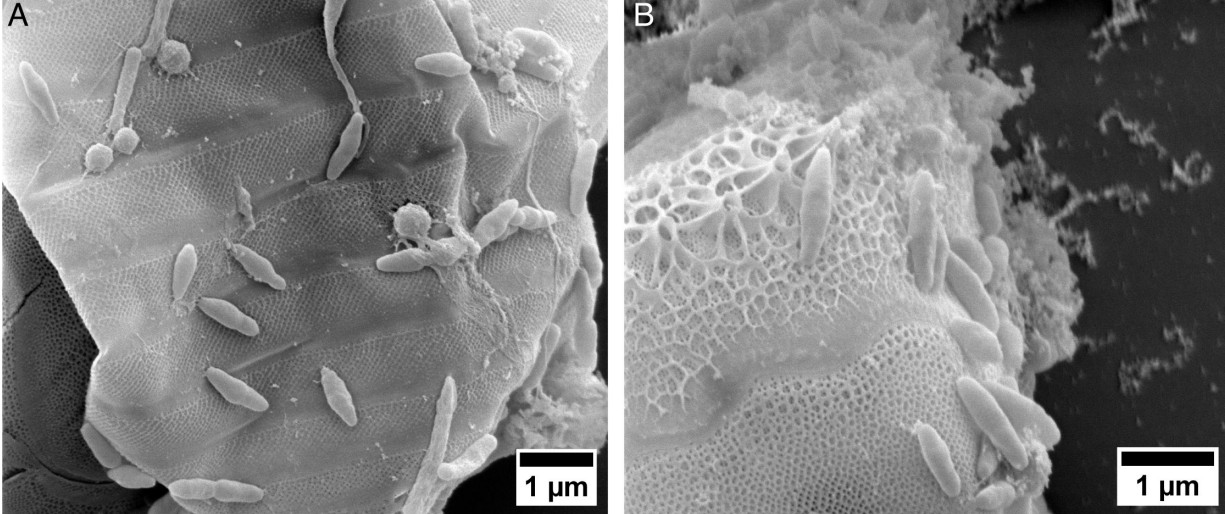

**FIG 1** SEM images of bacteria colonizing the surface of *C. weissflogii* cells. Different morphologies are visible (A) along with the concentration of cells in areas with unknown polymeric substances (B).

## Bacterial community composition and diversity

The bacterial communities analyzed by 16S rRNA gene amplicon sequencing contained a total of 3,859 ASVs across all 68 samples after removal of contaminant ASVs, chloroplasts, and mitochondria. The highest richness was observed at the beginning of the experiment with a mean Chao1 richness of 909 (day 0). Over time, richness decreased in all cultures to a mean of 201 (Chao1, day 49; Fig. S4).

The microbiome inoculum (i.e., filtered through 3 µm filters) was not statistically different from the microbiomes of the raw seawater before filtration, both in terms of composition, Chao1 richness (analysis of variance [ANOVA], *P*-value = 0.443), and Shannon diversity (ANOVA, *P*-value = 0.635), indicating that only few low abundance bacterial taxa were lost during the 3 µm filtration step (Fig. S5). The inoculum microbiome had a high richness (1537, Chao1) along with high diversity (3.97, Shannon's diversity index) (Table S2), and contained many taxonomic groups, which decreased rapidly in the assembling microbiomes, including Ilumatobacteraceae, Sporicthyaceae, SAR116 clade members, and SAR86 clade members. After 7 days, the microalgal microbiomes differentiated strongly from the initial microbiomes, with divergence based on the host algal species (Fig. 2A) explaining 22.5% of the total variation in composition at day 7 (PERMANOVA, $R^2$ = 0.225, $P$ < 0.001, perm. = 999). Particularly the microbiomes associated with *I. galbana* differed from those of *T. suecica* and *C. weissflogii* (Fig. 2B). Over the course of the experiment, the microbiomes diverged further according to the algal host species, and also increasingly over time according to each biological replicate culture. A PERMANOVA across all time points during the assembly indicated that host species explained 11% of the total variation in microbiome compositions, whereas time explained 3.4%, and fraction (A or FL) merely 2.9% (all $P$ < 0.001, perm. = 999).

In general, *C. weissflogii* microbiomes had high richness and diversity with a mean Chao1 of 433 and Shannon's diversity index of 3.23 across all time points, compared to *I. galbana* and *T. suecica* microbiomes with mean Chao1 of 278 and 374, and mean Shannon's diversity index of 2.56 and 2.71, respectively (Table S2). Some of the most abundant families in the microbiomes of *I. galbana* were Flavobacteriaceae, Cryomorphaceae, Devosiaceae, Cyclobacteraceae, and Kordiimonadaceae, whereas *T. suecica* microbiomes were dominated mainly by Rhodobacteraceae, but also Cryomorphaceae, Flavobacteriaceae, and Rhizobiaceae. *C. weissflogii* microbiomes also harbored high abundances of Rhodobacteraceae, Cryomorphaceae, and Rhizobiaceae, and additionally Cyanobiaceae and Rubinisphaeraceae (Fig. 3 and 4).

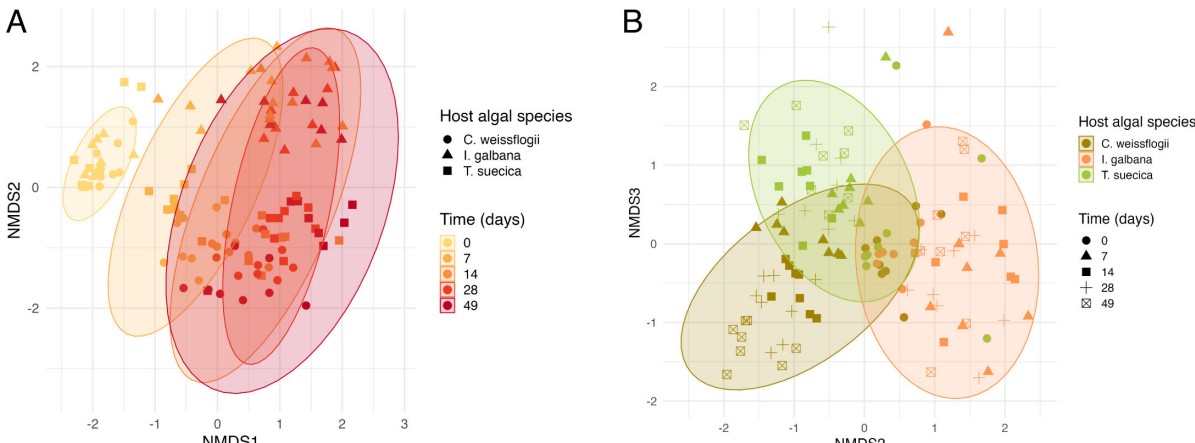

**FIG 2** NMDS ordinations of microbiome compositions associated with the three host microalgae over time-based on Bray Curtis dissimilarity. (A) NMDS dimensions 1 and 2 of microbiomes associated with *C. weissflogii* (circles), *I. galbana* (triangles), and *T. suecica* (squares). The time of sampling of the microbiomes is represented by colors, i.e., day 0 (yellow), day 7 (light orange), day 14 (orange), day 28 (bright red), and day 49 (dark red). Ellipses represent 95% CIs of microbiomes sampled at each timepoint. (B) NMDS dimensions 2 and 3 of microbiome compositions associated with *C. weissflogii* (olive-brown), *I. galbana* (orange), and *T. suecica* (green). The time of sampling of the microbiomes is represented by shapes, i.e., day 0 (circles), day 7 (triangles), day 14 (squares), day 28 (plus), and day 49 (square with ×). Ellipses represent 95% CIs of microbiomes associated with a certain host alga. According to a PERMANOVA, host algal species explained 11% of the total variation in this data set, time explained 3.4%, and fraction (A or FL) 2.9% (all $P < 0.001$, perm. = 999).

A total of 14 ASVs were identified as differentially abundant by ANCOM-BC (66). An ASV identified as a *Loktanella* sp. was differentially abundant in *T. suecica* microbiomes along with a *Marinomonas* sp. ASVs identified as belonging to the NS11-12 marine group, NS10 marine group, *Marinobacter, Halioglobus,* and *Vibrio* were differentially abundant in *C. weissflogii* microbiomes. Finally, ASVs differentially abundant in *I. galbana* microbiomes were assigned to the NS10 marine group and the *Marinoscillum* genus.

## Functional potential of microbiomes

From shotgun sequencing of the microalgal microbiomes at day 49, between 42M and 84M raw reads were obtained for each of the samples. For *C. weissflogii* microbiomes, between 9.7% and 37.7% of raw reads mapped to the host genome, whereas between 61.7% and 88.9% of raw reads from *I. galbana* microbiomes mapped to the host genome, and between 39.3% and 84.7% of raw reads mapped to the host genome in *T. suecica* microbiomes. The metagenomic assemblies of the microbiomes contained 2,131–7,318 contigs for the *I. galbana* microbiomes with N50s of 54,381–197,413. *T. suecica* microbiome assemblies contained between 5,380–6,801 contigs with N50s of 28,477–94,500, whereas microbiome assemblies from *C. weissflogii* cultures contained 15,691–24,778 contigs with N50s of 16,575–30,403 (Table S3). Taxonomy of the metagenomics assemblies was similar to those assessed with 16S rRNA amplicon sequencing with Flavobacteriaceae members dominating the *I. galbana* microbiomes,

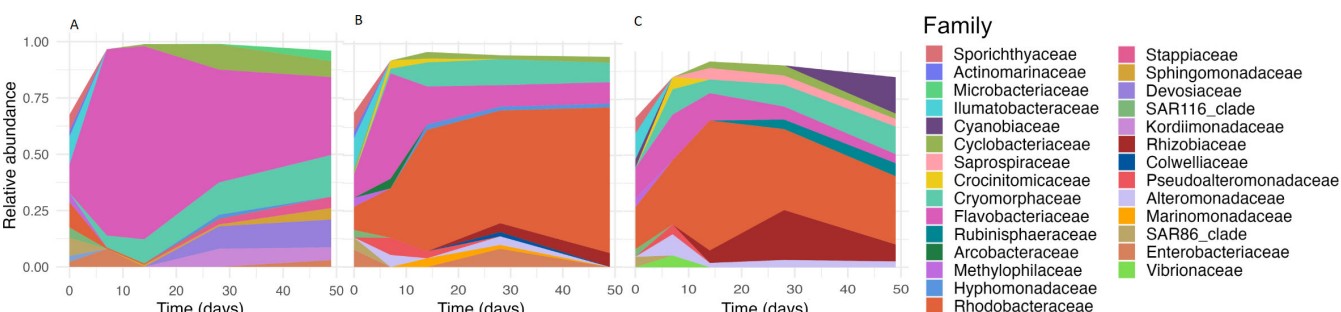

**FIG 3** Most abundant families in microbiomes associated with (A) *I. galbana*, (B) *T. suecica*, and (C) *C. weissflogii* over time.

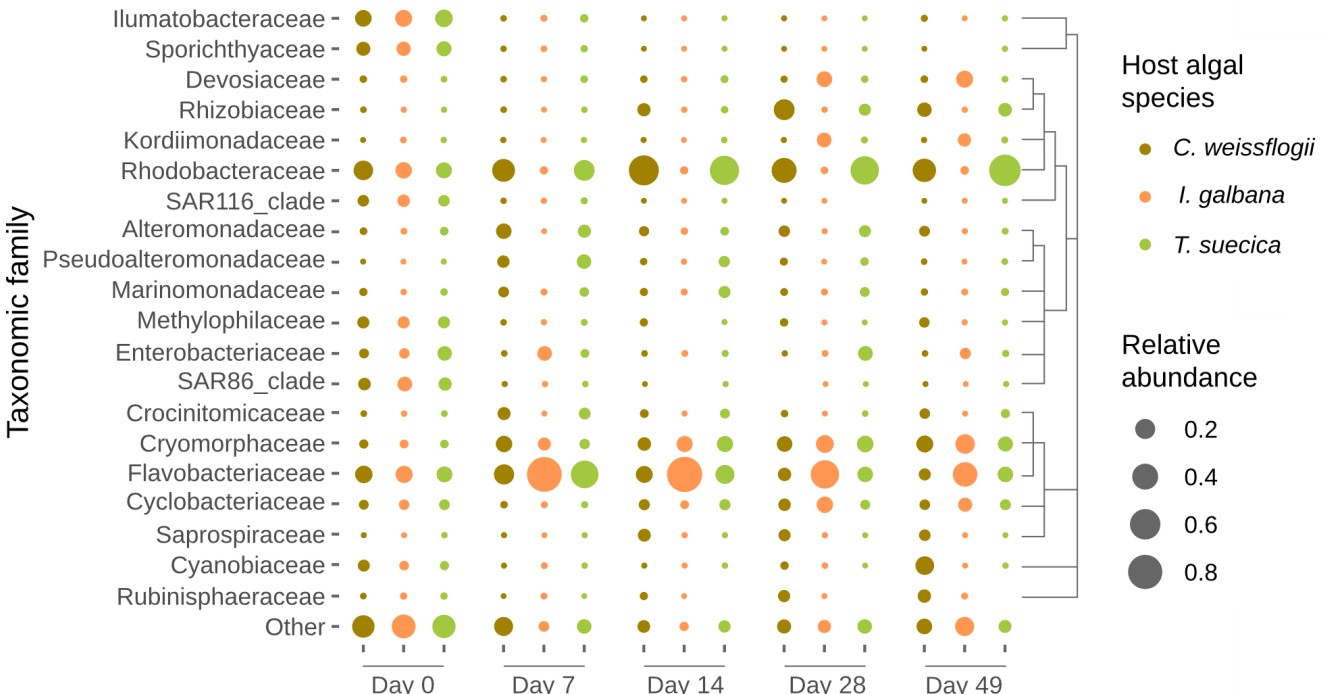

**FIG 4** Top 20 taxonomic families in microbiomes associated with *C. weissflogii* (olive-brown), *I. galbana* (orange), and *T. suecica* (green) across 49 days. The remaining families have been collapsed into "Other," and the relative abundance of each family is represented by the size of dots.

and Rhodobacteraceae members dominating the *T. suecica* and *C. weissflogii* microbiomes. No archeal or virus-like sequences were detected.

Based on the KOs assigned to genes present in the microbiome metagenomics assemblies (a total of 5,268 KOs), the functional potential of the microalgal microbiome assemblies was significantly different across different host microalgae (PERMANOVA, $R^2$ = 0.468, $P$ < 0.001, perm. = 999). While many abundant potential functions were present across microbiomes associated with all three host microalgae (Fig. 5 and 6A), some were present mostly or only in microbiomes associated with one or two of the microalgal host species (Fig. 5 and 6B). Predicted functions present in the majority of microbiomes associated with *C. weissflogii* were type I secretion, sulfur dioxygenase, NAD(P)H-quinone oxidoreductase, NAD-reducing hydrogenase, staphyloaxanthin biosynthesis, nitric oxide reduction, nitrite reduction, cytochrome b6/f complex, formaldehyde assimilation, retinal from apo-carotenals, and photosystem I. Photosystem II was also predicted to be present in microbiomes associated with *C. weissflogii*, but also in *I. galbana* microbiomes. RuBisCo, nostoxanthin synthesis, myxoxanthophyll synthesis, and the Calvin-Benson-Bassham cycle were predicted in both *I. galbana* and *C. weissflogii* microbiomes. Functions predicted to be present in both *T. suecica* and *C. weissflogii* microbiomes were type VI secretion, CP-lyases, dissimilatory nitrate reduction, nitrite oxidation, thiosulfate oxidation, dissimilatory sulfate reduction, sulfite dehydrogenase, dimethylamine/trimethylamine dehydrogenase, methanogenesis (via trimethylamine), anoxygenic type-II reaction center, V-type ATPase, and dimethylsulfoniopropionate (DMSP) demethylation. Several pathways potentially involved in inter-kingdom interactions such as chemotaxis, motility (flagellum), ferric iron transporters, and synthesis of vitamins thiamin, riboflavin, and cobalamin were present in microbiomes associated with all three host microalgae (Fig. 5).

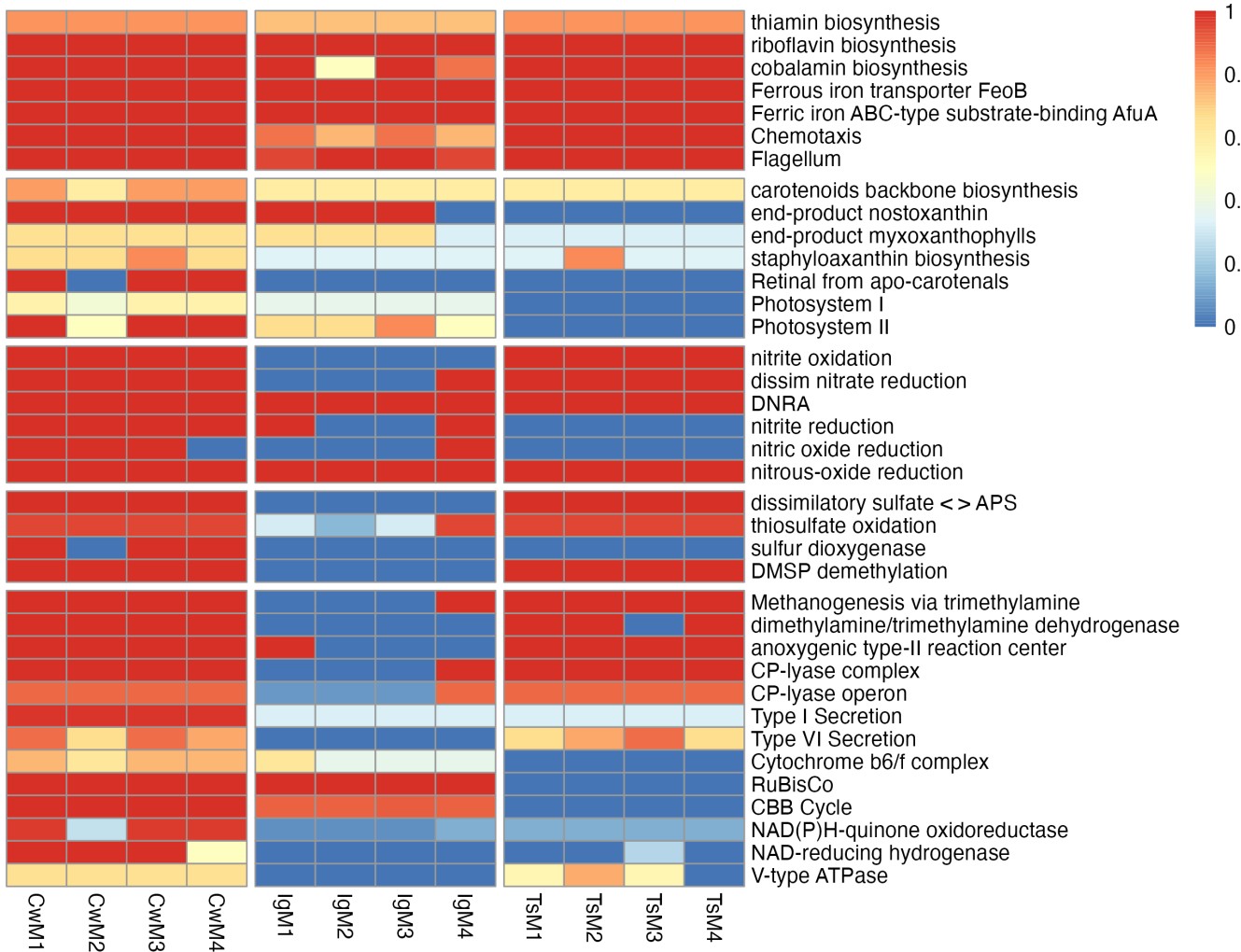

**FIG 5** KEGG pathways represented in microbiomes associated with *C. weissflogii* (CwM1-4 samples), *I. galbana* (IgM1-4 samples), and *T. suecica* (TsM1-4 samples). Pathways are grouped according to metabolic processes potentially involved in; from top to bottom: inter-kingdom interactions, pigments and photosystems, nitrogen cycling, sulfur cycling, miscellaneous. Colors correspond to the completeness of pathways. The figure was created based on output from KEGGdecoder (77).

CAZymes in the metagenomes were analyzed due to the potential variability in carbohydrate exudates among different host algae, which may consequently influence the CAZyme profiles of their associated microbiomes. The analysis found a total of 248 CAZymes, of which there were many glycosyl transferases (GTs) in all microbiomes, but especially in those associated with *C. weissflogii* and *T. suecica* (54.4% and 51.9% of total CAZymes, respectively) compared to those associated with *I. galbana* (41.7%; Fig. 6C). The metagenomes also contained many glycoside hydrolases (GHs), which made up 31.0%, 42.3%, and 33.4% of total CAZymes in microbiomes associated with *C. weissflogii, I. galbana,* and *T. suecica,* respectively. Carbohydrate esterases (CEs), auxiliary activity enzymes (AAs), carbohydrate-binding modules (CBMs), and polysaccharide lyases (PLs) were much less abundant in the assembled microbiomes, however, *T. suecica* microbiomes contained fewer CEs and more AAs, relatively, compared to microbiomes associated with *I. galbana* and *C. weissflogii,* and *I. galbana* microbiomes contained more PLs than microbiomes associated with the other two host algae. As such, the CAZymes breakdown was different for each microalgal host with ~50% of the overall variation in CAZymes compositions explained by different algal hosts (PERMANOVA, $R^2 = 0.495$, $P < 0.001$, perm. = 999; Fig. 6D).

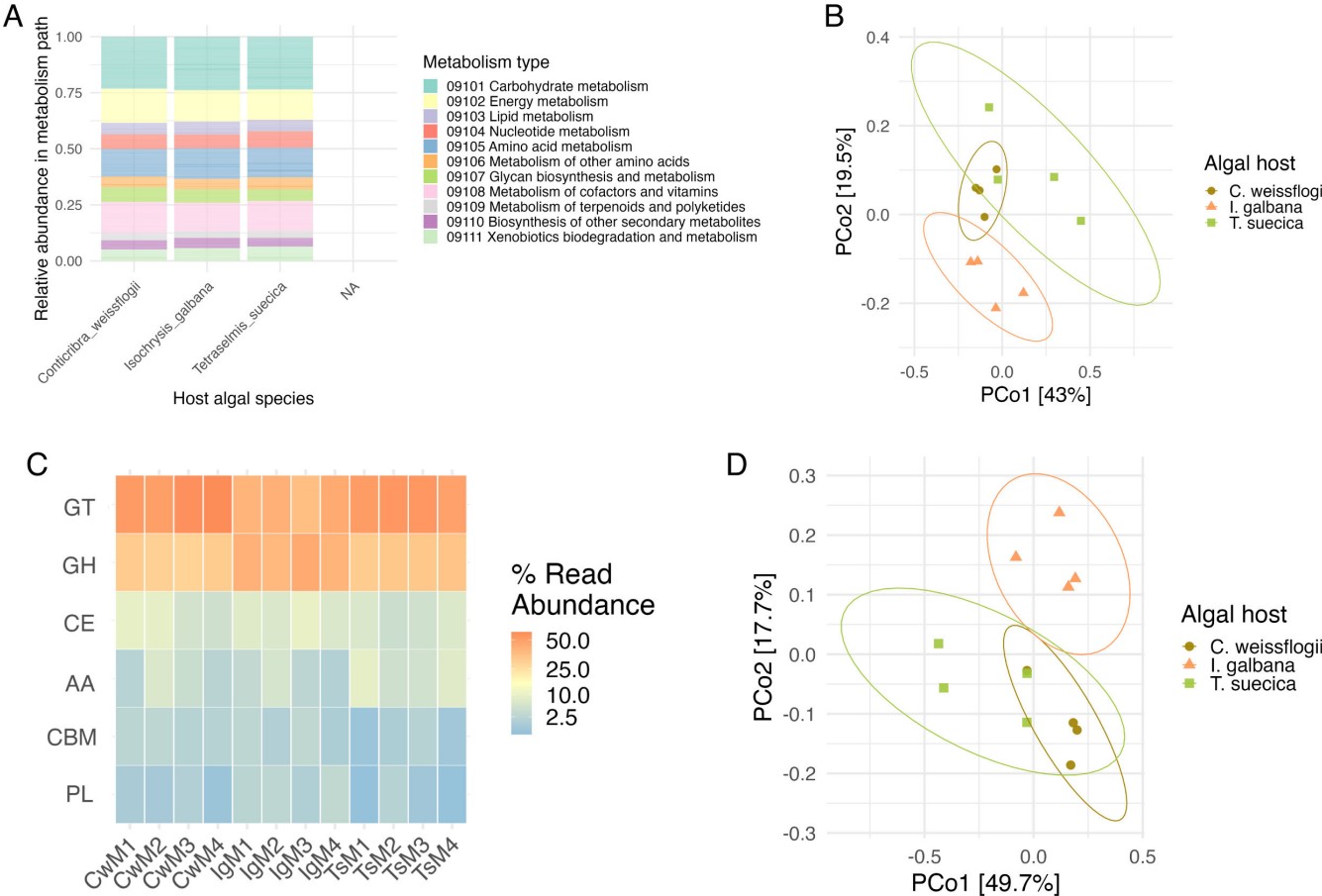

**FIG 6** Overview of potential functionality profiles of microbiomes associated with the three microalgal hosts. (A) Relative abundances of types of metabolisms as assigned by KOs in microbiomes associated with *C. weissflogii* (left), *I. galbana* (middle), and *T. suecica* (right). (B) Ordination of KOs in microbiomes associated with *C. weissflogii* (olive-brown), *I. galbana* (orange), and *T. suecica* (green). Ordination is based on Bray Curtis dissimilarity, and ellipses represent 90% of CIs for functional compositions of microbiomes associated with each algal host. (C) Heatmap of CAZyme classes in relative abundances (%) in microbiomes associated with *C. weissflogii* (samples denoted "Cw"), *I. galbana* (samples denoted "Ig"), and *T. suecica* (samples denoted "Ts"). Classes represented are glycosyl transferases (GT), glycoside hydrolases (GH), carbohydrate esterases (CE), auxiliary activities (AA), carbohydrate-binding modules (CBM), and polysaccharide lyases (PL). (D) Ordination of CAZymes compositions in microbiomes associated with *C. weissflogii* (olive-brown), *I. galbana* (orange), and *T. suecica* (green). Ordination is based on Bray Curtis dissimilarity, and ellipses represent 90% CIs of CAZymes compositions associated with each host alga.

## DISCUSSION

The phycosphere microbiomes assessed were significantly impacted by microalgal host species in terms of microbiome compositions (Fig. 2, PERMANOVA, $R^2 = 0.11$, $P < 0.001$, perm. = 999) and potential functionality (Fig. 6B, PERMANOVA, $R^2 = 0.468$, $P < 0.001$, perm. = 999). A main shift in microbiome compositions according to host species happened during the first 7 days of the experiment, however, microbiomes continued assembling and developing across the 49 days of the experiment (Fig. 3). It should be noted that the microbiomes assessed here developed under constant illumination conditions, which could affect microbiome compositions and activities.

Culturable bacterial counts of *I. galbana* and *T. suecica* increased over time when exposed to the same natural seawater microbiomes, whereas they decreased slightly toward the end of the experiment for *C. weissflogii* cultures (Fig. S1). While we do not have absolute bacterial counts for the whole experiment, we believe this increase in *I. galbana* and *T. suecica* cultures is due to a selection for heterotrophic bacteria thriving on the algal exudates and being capable of growing on marine agar. The decrease in culturable bacterial counts in *C. weissflogii* cultures toward the end of the experiment could be due to the aggregation observed and enhanced bacterial colonization

of these aggregates. It should be noted that with the methodology employed here, particle-attached bacteria in the collected seawater were possibly lost in the inoculum microbiomes introduced to the microalgal hosts due to being filtered out with larger particles in the 3 µm filtration step. In the North Sea close to Helgoland, Heins and Harder (78) observed a bacterial cultivability of 0.7%, which has a similar environment to the origin of the microbiome inoculum obtained here at Øresund and could indicate a similar cultivability. In the study by Heins and Harder, bacteria that were particle-associated had a higher cultivability than free-living bacteria, which could explain the increasing cultivability over time observed in the present study. Sanz-Sáez et al. (79) found a higher cultivability of deep ocean heterotrophic bacteria compared to those in the photic and mesopelagic zones, and the authors attribute this to deep ocean bacteria being dual-lifestyle bacteria which have attached to sinking particles and then become abundant in the deep ocean compared to the photic zone. Since phytoplankton form the basis of marine snow particles (80–82), it is likely that some of these bacteria could have been microalga-associated. Additionally, in the present study, families Ilumatobacteraceae, Sporicthyaceae, SAR116 clade members, and SAR86 clade members were among the most abundant taxonomic families in the natural seawater microbiome inoculum, but rapidly decreased in abundance and even disappeared in some algal cultures (Fig. 4). We did not measure the cultivability of bacteria in the seawater collected for this study, however, these families and clades are known to harbor bacteria that are recalcitrant to cultivation under laboratory conditions (83), making it likely that the environment of laboratory microalgal cultures selected for cultivable bacteria.

A range of taxonomic families were observed to replace recalcitrant-to-cultivation (under standard laboratory conditions) taxonomic groups in the microalgal microbiomes, such as members of the Rhodobacteraceae, Flavobacteriaceae, Cryptomorphaceae, Cyclobacteriaceae, and Rhizobiaceae families (Fig. 4). The latter has previously been observed in microbiomes associated with the haptophyte *Phaeocystis* (84), however, in the present study, this family was mostly associated with the diatom and the chlorophyte. This concurs with the findings of Steinrücken et al. (37), who found Rhizobiaceae in microbiomes associated with *T. suecica* and the diatom *Phaeodactylum*. The presence of this family is interesting given the known plant-growth-promoting effects of some Rhizobiaceae bacteria (85). Mars Brisbin et al. (84) also found the Alteromonadaceae family to be abundant in *Phaeocystis* microbiomes, which other studies have also identified as a family commonly occurring in microalgal microbiomes (31, 32, 37, 39). In the present study, Alteromonadaceae was not a highly abundant family, however, an ASV characterized as a *Marinobacter* sp. was differentially abundant in *C. weissflogii* microbiomes. Several other studies have documented interactions between *Marinobacter* spp. and diatoms, where the bacterium increases iron bioavailability to its host or induces aggregation (86–89). While unknown if this was the case in the present study, aggregation of the diatom was observed exclusively in cultures with the microbiome inoculum.

Rhodobacteraceae and Flavobacteriaceae are families that are commonly associated with phytoplankton blooms and microalgal microbiomes (2, 40, 90). Rhodobacteraceae have been found in association with several species of diatoms, including *Phaeodactylum* and *Thalassiosira* (31, 32, 91). They are generally capable of degrading low-molecular-weight carbohydrates and are thought to capitalize on partial degradation of algal exudates by other bacteria in the phycosphere (40, 42). In a study exploring the microbiomes associated with *Thalassiosira rotula,* the most commonly found ASV in samples from across the Pacific, Indian, and Atlantic oceans (present in 59% of samples) was assigned as a *Loktanella* sp. (31). A *Loktanella* sp. was a core member of the microbiomes of *T. suecica* and *C. weissflogii* in the present study as well, demonstrating a possible preference of this ecological niche by *Loktanella* spp. Other Rhodobacteraceae members, *Ruegeria* and *Rhodobacter* spp., have not only been associated with the chlorophyte *Chlorella* through transcriptomics analyses but correlated with a faster growth rate of the host (92). While the growth rate of *T. suecica* with its microbiome was not higher than that of axenic cultures in the present study (Fig. S2), it is possible

that a growth-enhancing effect of some Rhodobacteraceae members could be observed in co-culture. Another study provided evidence that another chlorophyte, *Ostreococcus* sp., was beneficially affected in terms of culture stability by two *Roseovarius* strains, potentially through cobalamin synthesis (30).

In addition to Rhodobacteraceae, members of the Flavobacteriaceae family were abundant across the microalgal microbiomes, but especially in association with *I. galbana* (Fig. 3). Members of this family have repeatedly been observed in correlation with phytoplankton blooms as well as laboratory-based microalgal microbiomes (29, 32, 36, 40, 42). Flavobacteriaceae generally carry the capacity to degrade high-molecular-weight (HMW) carbohydrates, and it has previously been hypothesized that this trait is the driving factor behind their frequent association with phytoplankton blooms (36, 42, 93, 94). *Polaribacter* spp. have been observed in several studies as associated with both diatoms and the haptophyte *E. huxleyi* (29, 36), and it was also a core member of the microbiomes in the present study, being present in over 80% of all samples. Several other Flavobacteriaceae genera were also core constituents of the microalgal microbiomes studied here, including the NS3a marine group and *Muricauda*. While associated with all three host algae in the present study, the NS3a marine group has previously been observed in high abundances during a diatom and *Phaeocystis* bloom (95), and *Muricauda* has been observed in *T. suecica* microbiomes, where it has been reported to enhance the growth of the host (96, 97).

While some bacterial families were abundant in microbiomes across microalgal hosts, the microbiome compositions did differ according to microalgal host (Fig. 2). Hence, "archetypal phytoplankton-associated bacteria" (40) may exist at the taxonomic level of family and higher. However, microbiome compositions on the ASV level tend to be host-specific, as also observed here. Host specificity of microalgal microbiomes has been observed in several studies spanning a breadth of hosts as phylogenetically different as the ones selected in the present study (31–34, 84, 98–100). However, some studies identified other effects influencing microalgal microbiome composition such as the origin of the initial inoculum, culture conditions, and stochastic effects (36, 37, 91), and yet other studies have found no significant effect of microalgal host on microbiome composition (38, 101). It is evident that the mechanisms of microalgal microbiome assembly and composition are still not fully understood, however, a majority of studies, including this work, point toward microalgal host species (perhaps strain) being one of the determining factors.

In addition to microalgal microbiomes being host-specific in terms of taxonomic composition, the present study also found host-specificity in terms of potential functionality as assigned by KOs. Overall functional groups appear more conserved than taxonomic composition at a similarly high level (Fig. 6A), which corresponds to the trend observed in human-associated microbiomes (43, 44). At a lower level though, both KO and CAZymes analysis showed a highly host-specific composition (Fig. 6B and D). Many different pathways were present in one or two replicate microbiomes associated with each microalgal host, however, only pathways that were detected in the majority (≥3) of replicate microbiomes with a specific microalgal host were considered here, as they are likely retained due to an importance for proliferation of microbiome members in association with the microalgal host. Several pathways related to photosynthesis and pigment production were prevalent in *I. galbana* and *C. weissflogii* microbiomes, where cyanobacteria were present. Interestingly, in *C. weissflogii* microbiomes, several pathways that are commonly related to anaerobic conditions were present, including methanogenesis-related pathways, and all steps involved in the denitrification pathway (Fig. 5). While the phycosphere is generally considered an aerobic environment, aggregation of *C. weissflogii* cells was observed throughout the experiment, which could cause anaerobic microenvironments to form inside aggregates (102, 103). We speculate that this aggregation of *C. weissflogii* creates selective pressure for bacteria capable of denitrification and other anaerobic metabolisms beneficial in anoxic microenvironments. In addition, methylamines have been identified as a source of cross-feeding between

Rhodobacteraceae capable of their degradation and diatoms taking up the ammonium resulting from degradation (104, 105), and denitrification pathway genes have previously been found in metagenome-assembled genomes from Rhodobacteraceae members in microbiomes associated with the eustigmatophyte *Microchloropsis salina* (98). Indeed, several Rhodobacteraceae members including *Loktanella* and *Lentibacter* harbored genes encoding methanogenesis via trimethylamine pathways, indicating that similar cross-feeding mechanisms may be taking place in this system.

Nutrients are likely being exchanged between microalgae and their microbiomes, according to previous studies as well as this study (3, 18, 86, 105). In addition to nitrogen, carbon in different forms supports the growth of both the microalgal host and their associated microbiomes. Apart from the members performing photosynthesis, most of the microbiome members are likely to obtain carbon in the form of organic exudates from the host microalgae. Eigemann et al. (106) found phytoplankton-released dissolved organic matter from two different hosts, a diatom and a cyanobacterium, to shape bacterial communities differently based on the producer species. This corresponds to the effect seen in the current study, where CAZymes profiles were significantly different according to microalgal host species (Fig. 6C and D). The CAZymes profiles also reveal a pattern similarly observed in the microbiome composition data; the microbiomes associated with *I. galbana* diverge significantly from those of *T. suecica* and *C. weissflogii*. While *T. suecica* and *C. weissflogii* microbiomes and CAZymes profiles are significantly different from each other, they share greater similarities when compared to the *I. galbana* microbiomes. We hypothesize that this could be due to *I. galbana* being a bacterivorous mixotroph, which could influence carbon availability in these systems. In comparison, *C. weissflogii* is a strict photoautotroph and while *T. suecica* can in some cases assimilate organic carbon, it is not bacterivorous so far (107–109). While this remains speculation, it is possible that the bacterivorous nature of *I. galbana* alters the composition of carbohydrate exudates, possibly providing less bioavailable carbohydrates, which could cause the prevalence of Flavobacteriaceae in these microbiomes due to their ability to degrade HMW carbohydrates. Interestingly, another study has previously observed positive associations between another bacterivorous mixotroph, *Micromonas,* and Flavobacteriaceae (110). In the present study, the functional profiles of the microalgal microbiomes are only partly explained by the trophic modes of their respective hosts (PERMANOVA, $R^2 = 0.163$, $P = 0.005$, perm. = 999), whereas the CAZymes profiles are explained to a higher degree by the trophic mode of the microalgal host (PERMANOVA, $R^2 = 0.357$, $P = 0.003$, perm. = 999). It is also possible that *I. galbana* feeds selectively on some bacterial taxa or sizes of bacteria as seen in *Rhodomonas* sp. (111).

## Conclusion

Our study demonstrates that the microbiomes of three marine host microalgae, *I. galbana, T. suecica,* and *C. weissflogii* diverged according to host microalgal species across a 49-day experiment under constant illumination, which was initiated with the same natural seawater microbiome. The shift was evident within the first 7 days, which highlights a significant host specificity in microalgal microbiome compositions. Rhodobacteraceae and Flavobacteriaceae families dominated the microalgal microbiomes, confirming their common association with microalgae and phytoplankton blooms. Intriguingly, the microbiomes of *I. galbana* exhibited pronounced dissimilarity from the microbiomes of *T. suecica* and *C. weissflogii*, which were mostly dominated by Rhodobacteraceae, whereas *I. galbana* microbiomes were mostly dominated by Flavobacteriaceae. This is particularly interesting given the bacterivorous mixotrophic nature of *I. galbana*, which could alter the composition of carbohydrate exudates available to its associated microbiome.

The functional potential of the microbiomes varied according to microalgal host species, including variations in the CAZymes profiles. These functional divergences could have critical implications for the health and ecological roles of the host microalgae such as aggregation and sinking behavior (88, 89), and hence should be considered

in future studies. Additionally, our metagenomics assembly data revealed that all steps in denitrification along with other anaerobic pathways were present in most replicate microbiomes (one replicate missing nitric oxide reduction genes) of *C. weissflogii* while not consistently present in microbiomes associated with the two other host microalgae. We speculate that this may be due to anoxic microenvironments forming within aggregates formed by the diatom.

Our findings underline the significant influence of host microalgal species in shaping the taxonomic and functional composition of associated microbiomes. These results will aid in further understanding microalgal microbiomes across phyla and their ecological implications as well as the potential engineering of said microbiomes for algal health and biotechnological purposes.

## ACKNOWLEDGMENTS

Robert Murphy, Mikael Lenz Strube, and Pernille Kjersgaard Bech are thanked for discussions on bioinformatics and visualizations, Martin Gachenot for advice on flow cytometry, and Nathalie Nina Suhr Eiris Henriksen for help with sampling.

This project was funded by the Novo Nordisk Foundation (NNF20OC0064249), which also provided equipment funding (NNF19OC0055625).

## AUTHOR AFFILIATIONS

[1]Department of Biotechnology and Biomedicine, Technical University of Denmark, Kgs. Lyngby, Denmark
[2]DTU Nanolab, National Center for Nano Fabrication and Characterization, Technical University of Denmark, Kgs. Lyngby, Denmark
[3]Department of Health Technology, Technical University of Denmark, Kgs. Lyngby, Denmark

## PRESENT ADDRESS

Eva C. Sonnenschein, Swansea University, School of Biosciences, Geography and Physics, Singleton Park, Swansea, Wales, United Kingdom

## AUTHOR ORCIDs

Line Roager  http://orcid.org/0000-0002-7033-7309
Mikkel Bentzon-Tilia  http://orcid.org/0000-0002-7888-9845
Eva C. Sonnenschein  http://orcid.org/0000-0001-6959-5100
Lone Gram  http://orcid.org/0000-0002-1076-5723

## FUNDING

| Funder | Grant(s) | Author(s) |
| --- | --- | --- |
| Novo Nordisk Fonden (NNF) | NNF20OC0064249,NNF19OC0055625 | Line Roager |
| | | Eva C. Sonnenschein |
| | | Lone Gram |

## DATA AVAILABILITY

Raw 16S sequencing data is available on NCBI with BioProject ID PRJNA1031939 and raw metagenomics sequencing data on BioProject ID PRJNA1034224. Scripts for metagenomics data analysis can be found on github.com/LineRoager at https://github.com/lineroager/phycosphere_microbiomes_scripts.

## ADDITIONAL FILES

The following material is available online.

## Supplemental Material

**Supplemental figures (mSystems00583-24-s0001.pdf).** Fig. S1 to S5.
**Supplemental tables (mSystems00583-24-s0002.docx).** Tables S1 to S3.

## Open Peer Review

**PEER REVIEW HISTORY (review-history.pdf).** An accounting of the reviewer comments and feedback.

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
