## [Reviewer comments · mSystems]

Impact of host species on assembly, composition, and functional profiles of phycosphere microbiomes

Line Roager, Paul Kempen, Mikkel Bentzon-Tilia, Eva Sonnenschein, and Lone Gram

Corresponding Author(s): Lone Gram, Danmarks Tekniske Universitet Institut for Bioteknologi og Biomedicin

Review Timeline:

Submission Date:	April 24, 2024
Editorial Decision:	May 24, 2024
Revision Received:	June 28, 2024
Accepted:	July 1, 2024

Editor: Hans Bernstein

Reviewer(s): Disclosure of reviewer identity is with reference to reviewer comments included in decision letter(s). The following individuals involved in review of your submission have agreed to reveal their identity: Hannah Doris Schweitzer (Reviewer #1)

Transaction Report:

DOI: <https://doi.org/10.1128/msystems.00583-24>

Re: mSystems00583-24 (Impact of host species on assembly, composition, and functional profiles of phycosphere microbiomes)

Dear Prof. Lone Gram:

The original concerns were well addressed. One of the reviewers had just a few more comments that should be easy to satisfy and will improve the paper.

Revision Guidelines

Sincerely,
Hans Bernstein
Editor
mSystems

Reviewer #1 (Comments for the Author):

The authors provide a dataset consisting of 16S and metagenomic sequencing from laboratory cultures consisting of 3 different algae strains that were inoculated with a natural mixed community from seawater microbiome. The three algae strains used were *I. galbana*, *T. suecica*, and *C. weissflogii*. This dataset is important as it utilizes bacteria collected from seawater and lets the

microbiome for each algal strain be selected for over the course of 49 days in the laboratory. The authors also looked at the functional potential of the selected microbiomes for each of the algae after 49 days.

The information provided in this manuscript will add value to the research community. There are a few instances where the authors should provide clarification in their interpretations:

The authors say that the work presented here shows a deterministic role of the microalgal host species. I am curious if there were any statistical measurements of this stochasticity? The authors state this is deterministic at line 41 and 555.

Why are the *T. suecica* and *I. galbana* strains kept at a more similar concentration (10^5) compared to the *C. weissflogii* (10^3)? This seems like it could play a role in the differences seen between these samples from flocculation to genes expressed. Will the bacteria be able to come into contact with the algae at a similar rate if the algae are less dense in one culture?

Since the substance on the outside of the algae is identified by comparing the axenic SEM image to the microbiome SEM image it would be important for the readers to be able to see these in the same magnification. Could they both be presented in $1\mu\text{m}$ so we can confidently see the difference?

Line 301 - How do the authors know this substance is from the algae?

Line 302 - Did the authors do any counts are quantifiable measurements to be able to say more bacteria were on one algae strain compared to the other?

Why are the KOs presented showing the differences in the replicates but then when looking at the CAZymes they are just showing the difference between the 3 algae strains. This should remain consistent. The authors should either keep the gene abundance information separate or combine it.

Line 522 - The authors say it is evident that nutrients are being exchanged in this paper, how is it evident? Where do the authors show this occurring? The authors only present functional potential.

Minor:

All of the references to the tables and figures are not capitalized. I believe it is most often presented in publication like (Fig. 1A) and (Table 1). Could this be capitalized instead?

Line 174 - Do the authors only quantify *I. galbana* and *T. suecica* based on FSC and chlorophyll? How do the authors quantify *C. weissflogii*?

Line 290 - These is crazy error in the algae cell counts. Is this something the authors would expect?

Line 325 - fraction (A or FL) - where is this in the figure? I don't see it labelled anywhere and am confused where it connects.

Line 334 - Remove 'were abundant families' Since it starts with 'were dominated mainly by' they are redundant.

Line 409-412 - This is just a supportive comment saying that this supports what the authors saw from SEM with more bacteria being attached to *C. weissflogii* (although provide some quantification). So it would make sense to have less bacteria in these culture counts.

Line 412 - The order of this sentence is confusing because it makes it sound like the authors both observed the same cultivability which the authors didn't measure. Maybe say something like: Heins & Harder (2022) observed a bacterial cultivability of 0.7% in the North Sea close to Helgoland, which has a similar environment to the origin of the microbiome inoculum obtained here at Øresund and could indicate a similar cultivability.

Line 448-480 - This is a really long paragraph. Maybe separate at line 465 : New paragraph starting at 'In addition to Rhodobacteraceae,...

Line 497 - This can be pointed out in the figure.

Line 536 - The authors say bacterivorous mixotrophy could influence carbon availability but then don't explain how until line 543. This is confusing and should be presented right after the point is presented instead of having a few sentences inbetween the point and the how.

Line 566 - Can the authors give some examples of how the health and ecological roles are impacted?

Line 569 - When the authors say 'most' can they give an quantifiable number of what most is?

Figure comments

It was hard to tell what figures corresponded to what based on how the file was downloaded but I did my best. (This is not the authors fault I believe just a comment saying that if I have a confusion on the figures it might be how I was able to see the file) In the discussion section try to point out where the authors points are shown in the figure. I tried to show a few places this can be done above.

The figures seem to go out of order at one point: The order presented for Figure 6 is 6A, 6C, 6B, 6D). Should stay in the correct order when presented in the paper.

Figure 5 and 6 - Why is figure 5 in replicates of 4 and figure 6 is not in replicates. The information is from the same samples correct?

Figure S2. Be consistent. If call it axenic or control keep consistent in the figure legend vs the actual labels in the figure

Figure S3 and Figure 1 should be the same magnification

The authors provide a dataset consisting of 16S and metagenomic sequencing from laboratory cultures consisting of 3 different algae strains that were inoculated with a natural mixed community from seawater microbiome. The three algae strains used were *I. galbana*, *T. suecica*, and *C. weissflogii*. This dataset is important as it utilizes bacteria collected from seawater and lets the microbiome for each algal strain be selected for over the course of 49 days in the laboratory. The authors also looked at the functional potential of the selected microbiomes for each of the algae after 49 days.

The information provided in this manuscript will add value to the research community. There are a few instances where the authors should provide clarification in their interpretations:

The authors say that the work presented here shows a deterministic role of the microalgal host species. I am curious if there were any statistical measurements of this stochasticity? The authors state this is deterministic at line 41 and 555.

Why are the *T. suecica* and *I. galbana* strains kept at a more similar concentration (10^5) compared to the *C. weissflogii* (10^3)? This seems like it could play a role in the differences seen between these samples from flocculation to genes expressed. Will the bacteria be able to come into contact with the algae at a similar rate if the algae are less dense in one culture?

Since the substance on the outside of the algae is identified by comparing the axenic SEM image to the microbiome SEM image it would be important for the readers to be able to see these in the same magnification. Could they both be presented in $1\mu\text{m}$ so we can confidently see the difference?

Line 301 – How do the authors know this substance is from the algae?

Line 302 – Did the authors do any counts or quantifiable measurements to be able to say more bacteria were on one algae strain compared to the other?

Why are the KOs presented showing the differences in the replicates but then when looking at the CAZymes they are just showing the difference between the 3 algae strains. This should remain consistent. The authors should either keep the gene abundance information separate or combine it.

Line 522 – The authors say it is evident that nutrients are being exchanged in this paper, how is it evident? Where do the authors show this occurring? The authors only present functional potential.

Minor:

All of the references to the tables and figures are not capitalized. I believe it is most often presented in publication like (Fig. 1A) and (Table 1). Could this be capitalized instead?

Line 174 – Do the authors only quantify *I. galbana* and *T. suecica* based on FSC and chlorophyll? How do the authors quantify *C. weissflogii*?

Line 290 – There is a crazy error in the algae cell counts. Is this something the authors would expect?

Line 325 – fraction (A or FL) – where is this in the figure? I don't see it labelled anywhere and am confused where it connects.

Line 334 – Remove ‘were abundant families’ Since it starts with ‘were dominated mainly by’ they are redundant.

Line 409-412 – This is just a supportive comment saying that this supports what the authors saw from SEM with more bacteria being attached to *C. weissflogii* (although provide some quantification). So it would make sense to have less bacteria in these culture counts.

Line 412 – The order of this sentence is confusing because it makes it sound like the authors both observed the same cultivability which the authors didn’t measure. Maybe say something like: Heins & Harder (2022) observed a bacterial cultivability of 0.7% in the North Sea close to Helgoland, which has a similar environment to the origin of the microbiome inoculum obtained here at Øresund and could indicate a similar cultivability.

Line 448-480 – This is a really long paragraph. Maybe separate at line 465 : New paragraph starting at ‘In addition to Rhodobacteraceae,...’

Line 497 – This can be pointed out in the figure.

Line 536 – The authors say bacterivorous mixotrophy could influence carbon availability but then don’t explain how until line 543. This is confusing and should be presented right after the point is presented instead of having a few sentences inbetween the point and the how.

Line 566 – Can the authors give some examples of how the health and ecological roles are impacted?

Line 569 – When the authors say ‘most’ can they give an quantifiable number of what most is?

Figure comments

It was hard to tell what figures corresponded to what based on how the file was downloaded but I did my best. (This is not the authors fault I believe just a comment saying that if I have a confusion on the figures it might be how I was able to see the file)

In the discussion section try to point out where the authors points are shown in the figure. I tried to show a few places this can be done above.

The figures seem to go out of order at one point: The order presented for Figure 6 is 6A, 6C, 6B, 6D). Should stay in the correct order when presented in the paper.

Figure 5 and 6 – Why is figure 5 in replicates of 4 and figure 6 is not in replicates. The information is from the same samples correct?

Figure S2. Be consistent. If call it axenic or control keep consistent in the figure legend vs the actual labels in the figure

Figure S3 and Figure 1 should be the same magnification

Reviewer #1 (Comments for the Author):

The authors provide a dataset consisting of 16S and metagenomic sequencing from laboratory cultures consisting of 3 different algae strains that were inoculated with a natural mixed community from seawater microbiome. The three algae strains used were *I. galbana*, *T. suecica*, and *C. weissflogii*. This dataset is important as it utilizes bacteria collected from seawater and lets the microbiome for each algal strain be selected for over the course of 49 days in the laboratory. The authors also looked at the functional potential of the selected microbiomes for each of the algae after 49 days.

The information provided in this manuscript will add value to the research community. There are a few instances where the authors should provide clarification in their interpretations:

The authors say that the work presented here shows a deterministic role of the microalgal host species. I am curious if there were any statistical measurements of this stochasticity? The authors state this is deterministic at line 41 and 555.

As there was no statistical measurement of deterministic effects other than the PERMANOVA confirming host species as an explanatory factor for microbiome compositions, we have changed this word to “significant” in both instances, now lines 41 and 561.

Why are the *T. suecica* and *I. galbana* strains kept at a more similar concentration (10^5) compared to the *C. weissflogii* (10^3)? This seems like it could play a role in the differences seen between these samples from flocculation to genes expressed. Will the bacteria be able to come into contact with the algae at a similar rate if the algae are less dense in one culture? *T. suecica* and *I. galbana* are algae of similar sizes (5-10 μm) and reach higher cell densities when grown in the lab compared to *C. weissflogii*, which are larger cells (>20 μm). Hence, for the algae to be able to grow unrestricted (i.e. up to 100-fold per week), we started the diatom cultures at lower cell densities. While it is true that this difference in density might affect both host and microbiomes, the larger cell size of *C. weissflogii* will compensate for the lower density to some extent. Additionally, since the difference observed between *I. galbana* and *T. suecica* microbiomes is just as significant as differences observed between *C. weissflogii* microbiomes and any of the other two, any such effect is not larger than other differences between the host species.

Since the substance on the outside of the algae is identified by comparing the axenic SEM image to the microbiome SEM image it would be important for the readers to be able to see these in the same magnification. Could they both be presented in 1 μm so we can confidently see the difference?

We have included different SEM images at the same magnification as those of the microbiome on *C. weissflogii* in figure S3.

Line 301 - How do the authors know this substance is from the algae?

This substance could indeed be from microbiome members as well. We have altered this line to include this uncertainty. Lines 301-302 now reads: “which we speculate is extracellular polymeric substances (EPS) produced by the alga or by microbiome members”.

Line 302 - Did the authors do any counts are quantifiable measurements to be able to say more bacteria were on one algae strain compared to the other?

No, this was not quantitatively assessed. We have included a comment about this in line 304.

Why are the KOs presented showing the differences in the replicates but then when looking at the CAZymes they are just showing the difference between the 3 algae strains. This should remain consistent. The authors should either keep the gene abundance information separate or combine it.

This is a good point. We have altered figure 6B to represent CAZymes according to separate replicates for consistency.

Line 522 - The authors say it is evident that nutrients are being exchanged in this paper, how is it evident? Where do the authors show this occurring? The authors only present functional potential.

This is true, and the sentence has been altered to indicate this. Line 528 now reads: "It is highly likely that...".

Minor:

All of the references to the tables and figures are not capitalized. I believe it is most often presented in publication like (Fig. 1A) and (Table 1). Could this be capitalized instead?

This has been changed throughout the manuscript.

Line 174 - Do the authors only quantify *I. galbana* and *T. suecica* based on FSC and chlorophyll? How do the authors quantify *C. weissflogii*?

C. weissflogii was quantified based on microscopy counts in Neubauer chambers. This information has been added in lines 175-176.

Line 290 - These is crazy error in the algae cell counts. Is this something the authors would expect?

Yes, this is similar to what we have previously seen, see e.g. Roager et al. (2024) "Antagonistic activity of *Phaeobacter piscinae* against the emerging fish pathogen *Vibrio crassostreae* in aquaculture feed algae". These deviations represent both biological variation as well as the fact that the algae grow exponentially and we are quantifying very small volumes of each culture.

Line 325 - fraction (A or FL) - where is this in the figure? I don't see it labelled anywhere and am confused where it connects.

This was not represented in the figure as fraction explained such a small part of the variation in the dataset, hence we only mention it here as part of the statistical results.

Line 334 - Remove 'were abundant families' Since it starts with 'were dominated mainly by' they are redundant.

This has been removed.

Line 409-412 - This is just a supportive comment saying that this supports what the authors saw

from SEM with more bacteria being attached to *C. weissflogii* (although provide some quantification). So it would make sense to have less bacteria in these culture counts.
This is a good point, which we have added in lines 411-413.

Line 412 - The order of this sentence is confusing because it makes it sound like the authors both observed the same cultivability which the authors didn't measure. Maybe say something like: Heins & Harder (2022) observed a bacterial cultivability of 0.7% in the North Sea close to Helgoland, which has a similar environment to the origin of the microbiome inoculum obtained here at Øresund and could indicate a similar cultivability.
This has been changed to the suggestion provided, now lines 416-419.

Line 448-480 - This is a really long paragraph. Maybe separate at line 465: New paragraph starting at 'In addition to Rhodobacteraceae,...'
This has been changed as suggested, the paragraph has been separated between lines 470 and 471.

Line 497 - This can be pointed out in the figure.
A reference to figure 6A has been added, now line 503.

Line 536 - The authors say bacterivorous mixotrophy could influence carbon availability but then don't explain how until line 543. This is confusing and should be presented right after the point is presented instead of having a few sentences inbetween the point and the how.
This paragraph has been reorganized according to this suggestion.

Line 566 - Can the authors give some examples of how the health and ecological roles are impacted?
An example has been added to lines 572-573.

Line 569 - When the authors say 'most' can they give an quantifiable number of what most is?
A comment about this has been added in lines 575-576.

Figure comments

It was hard to tell what figures corresponded to what based on how the file was downloaded but I did my best. (This is not the authors fault I believe just a comment saying that if I have a confusion on the figures it might be how I was able to see the file)
In the discussion section try to point out where the authors points are shown in the figure. I tried to show a few places this can be done above.
We have added more references to figures in the discussion section.

The figures seem to go out of order at one point: The order presented for Figure 6 is 6A, 6C, 6B, 6D). Should stay in the correct order when presented in the paper.
The order has been corrected both in the manuscript and in the figures, and figure 6C is now 6B and vice versa.

Figure 5 and 6 - Why is figure 5 in replicates of 4 and figure 6 is not in replicates. The information is from the same samples correct?

This has been adjusted as mentioned above.

Figure S2. Be consistent. If call it axenic or control keep consistent in the figure legend vs the actual labels in the figure

This has been changed.

Figure S3 and Figure 1 should be the same magnification

This has been changed as mentioned above.

Re: mSystems00583-24R1 (Impact of host species on assembly, composition, and functional profiles of phycosphere microbiomes)

Dear Prof. Lone Gram:

I am pleased by the constructive nature of comments and revisions made by the authors to address them.

Your manuscript has been accepted, and I am forwarding it to the ASM production staff for publication. Your paper will first be checked to make sure all elements meet the technical requirements. ASM staff will contact you if anything needs to be revised before copyediting and production can begin. Otherwise, you will be notified when your proofs are ready to be viewed.

Sincerely,
Hans Bernstein
Editor
mSystems